# Phonon transition across an isotopic interface

Ning Li[1,2,3,14], Ruochen Shi[1,2,14], Yifei Li[4,14], Ruishi Qi[1,2,5], Fachen Liu[1,2,3], Xiaowen Zhang[1,2], Zhetong Liu[1,2,3], Yuehui Li[1,2], Xiangdong Guo[6], Kaihui Liu [7,8], Ying Jiang [1,8,9], Xin-Zheng Li[7,9,10], Ji Chen [7,8,9] ✉, Lei Liu [4,9] ✉, En-Ge Wang [1,9,11,12] ✉ & Peng Gao [1,2,8,9,13] ✉

Isotopic mixtures result in distinct properties of materials such as thermal conductivity and nuclear process. However, the knowledge of isotopic interface remains largely unexplored mainly due to the challenges in atomic-scale isotopic identification. Here, using electron energy-loss spectroscopy in a scanning transmission electron microscope, we reveal momentum-transfer-dependent phonon behavior at the h–$^{10}$BN/h-$^{11}$BN isotope heterostructure with sub-unit-cell resolution. We find the phonons' energy changes gradually across the interface, featuring a wide transition regime. Phonons near the Brillouin zone center have a transition regime of ~3.34 nm, whereas phonons at the Brillouin zone boundary have a transition regime of ~1.66 nm. We propose that the isotope-induced charge effect at the interface accounts for the distinct delocalization behavior. Moreover, the variation of phonon energy between atom layers near the interface depends on both of momentum transfer and mass change. This study provides new insights into the isotopic effects in natural materials.

Isotopes of an element have the same number of protons, but different in numbers of neutrons and hence atomic mass[1], leading to different nuclei-governed properties, such as thermal conductivity[2–7], elasticity[8], and nuclear reactions[9,10]. The isotopic labeling is widely used as tracers in molecular labeling[11–13], chemical reactions[14–16], and radiometric dating[17–19]. Recent studies also report the isotope effect on chemcal/electronic properties, i.e. the superconductivty[20,21], the transmissivity[22], and the optical band gap[23]. These phenomena open a

new door for function design via isotope engineering, and also lead to essential questions for the naturally existing materials that commonly consist of isotopic mixture: are there new properties emerging at the isotope interface? For example, the isotope enriched materials have much enhanced thermal conductivity compared to natural ones[7,24,25], for which a well-accepted assumption is that the isotope mass disorder in natural materials disrupts the phonon transport accounting for the substantial reduction of thermal conductivity. Recently, effects of

[1]International Center for Quantum Materials, School of Physics, Peking University, 100871 Beijing, China. [2]Electron Microscopy Laboratory, School of Physics, Peking University, 100871 Beijing, China. [3]Academy for Advanced Interdisciplinary Studies, Peking University, 100871 Beijing, China. [4]School of Materials Science and Engineering, Peking University, 100871 Beijing, China. [5]Department of Physics, University of California at Berkeley, Berkeley, CA 94720, USA. [6]CAS Key Laboratory of Nanophotonic Materials and Devices, CAS Key Laboratory of Standardization and Measurement for Nanotechnology, CAS Center for Excellence in Nanoscience, National Center for Nanoscience and Technology, 100190 Beijing, China. [7]Institute of Condensed Matter and Material Physics, Frontiers Science Center for Nano-optoelectronics, School of Physics, Peking University, 100871 Beijing, China. [8]Collaborative Innovation Center of Quantum Matter, 100871 Beijing, China. [9]Interdisciplinary Institute of Light-Element Quantum Materials and Research Center for Light-Element Advanced Materials, Peking University, 100871 Beijing, China. [10]State Key Laboratory for Artificial Microstructure and Mesoscopic Physics, Peking University, 100871 Beijing, China. [11]Songshan Lake Materials Laboratory, 523808 Dongguan, China. [12]School of Physics, Shanghai University, 200444 Shanghai, China. [13]Hefei National Laboratory, 230088 Hefei, China. [14]These authors contributed equally: Ning Li, Ruochen Shi, Yifei Li. ✉e-mail: ji.chen@pku.edu.cn; l_liu@pku.edu.cn; egwang@pku.edu.cn; p-gao@pku.edu.cn

interface with one side of heterostructure isotopically purified, are studied. Phenomena like hyperfine isotope effects regulating interlayer electron-phonon coupling in heterostructures[26] and visualizing negative refraction of phonon polaritons[27] are reported. However, in the case of heterogeneous distribution of isotopes, the presence of isotope interfaces in a natural material makes it similar to the "superlattice" case. Under such a circumstance, the heat transport behavior is intuitively different from completely random ones[28]. Nevertheless, such knowledge on the isotope interface and possible effects has been rarely discussed before and thus largely unknown mainly due to the challenges in isotopic identification and property measurement at the atomic scale for their interfaces.

The commonly-used isotope analysis methods are based on vibration detection, such as Raman[2,3] and Infrared spectroscopy[29] with an energy resolution at the order of $1\,cm^{-1}$ to distinguish the phonons of isotopes, but they usually have a limited spatial resolution. Although the spatial resolution can be substantially improved by the tip-enhanced Raman spectroscopy[30] and scanning near-field optical microscopy[31], optical characterizations still lack the ability to resolve the atomic structure, and access to high-momentum phonons at the Brillouin zone (BZ) boundary due to the tiny momentum of photons. Recent advances in electron energy loss spectroscopy (EELS) in scanning transmission electron microscope (STEM) enable momentum-resolved vibrational measurements at nanometer/atomic scale[32–37], providing new opportunities to detect the isotopes distribution[38] e.g., H/D−O bonds[39], $^{13}C/^{12}C$−O bonds[40], and $^{13}C$−$^{13}C/^{12}C$−$^{12}C$ bonds[41].

Here, we fabricate atomically sharp h-$^{10}$BN/h-$^{11}$BN interface and characterize the lattice vibration properties at the interface with sub-unit-cell spatial resolution and different momentum transfer. We find that the out-of-plane optical phonon modes (dubbed as ZO) at the interface are not atomically sharp. Instead, they are delocalized, featured by a gradual transition across the interface. In addition, the

delocalization depends on the momentum transfer of the ZO mode, i.e., modes with small momentum transfer (ZO$_{low\ q}$ at BZ center) are delocalized ~3.34 nm, while the ones with large momentum transfer (ZO$_{high\ q}$ at BZ boundary) are delocalized ~1.66 nm. The different vibration amplitude of isotopes causes the different vibration dipole magnitude and thus momentum-dependent charge density at the interface, which is proposed to account for the different delocalization behavior. These two ZO modes also have different variation between atomic layers, i.e., the ZO$_{high\ q}$ is less sensitive to position variation (associated with the change of momentum transfer) as the phonon dispersion is flat at the BZ boundary. Furthermore, across the interface from the h-$^{10}$BN layer to h-$^{11}$BN layer, the vibrational energy of these two ZO phonons goes up first, then followed by a rapid drop. Such a complicated behavior can be qualitatively understood by the combination of changes in the scattering cross section and mass across the interface. The delocalization of phonon modes at an isotopic interface and the momentum-dependent behavior may affect the local thermal processes. These findings provide us with a new angle to understand the isotopic effects in natural materials and insights into tailoring property via isotopic engineering.

## Results

### Isotope identification at h-$^{11}$BN/h-$^{10}$BN interface

We grow large h-$^{11}$BN and h-$^{10}$BN isotope crystals (supplementary fig. 1) and then transfer a ~20-nm-thick h-$^{11}$BN and a ~100-nm-thick h-$^{10}$BN to SiO$_2$/Si substrate (methods) forming an atomically sharp h-$^{10}$BN/h-$^{11}$BN/a-SiO$_2$ (a-SiO$_2$: amorphous SiO$_2$) heterostructure[42]. The schematic of the assembled stack and HAADF image at the cross-section direction are shown in Fig. 1a, b. First-principles calculations based on density functional perturbation theory (DFPT) of the phonon dispersion and total phonon density of states (DOS) for h-$^{10}$BN and h-$^{11}$BN (supplementary fig. 2 and supplementary note 1) give an energy shift of ~5 meV

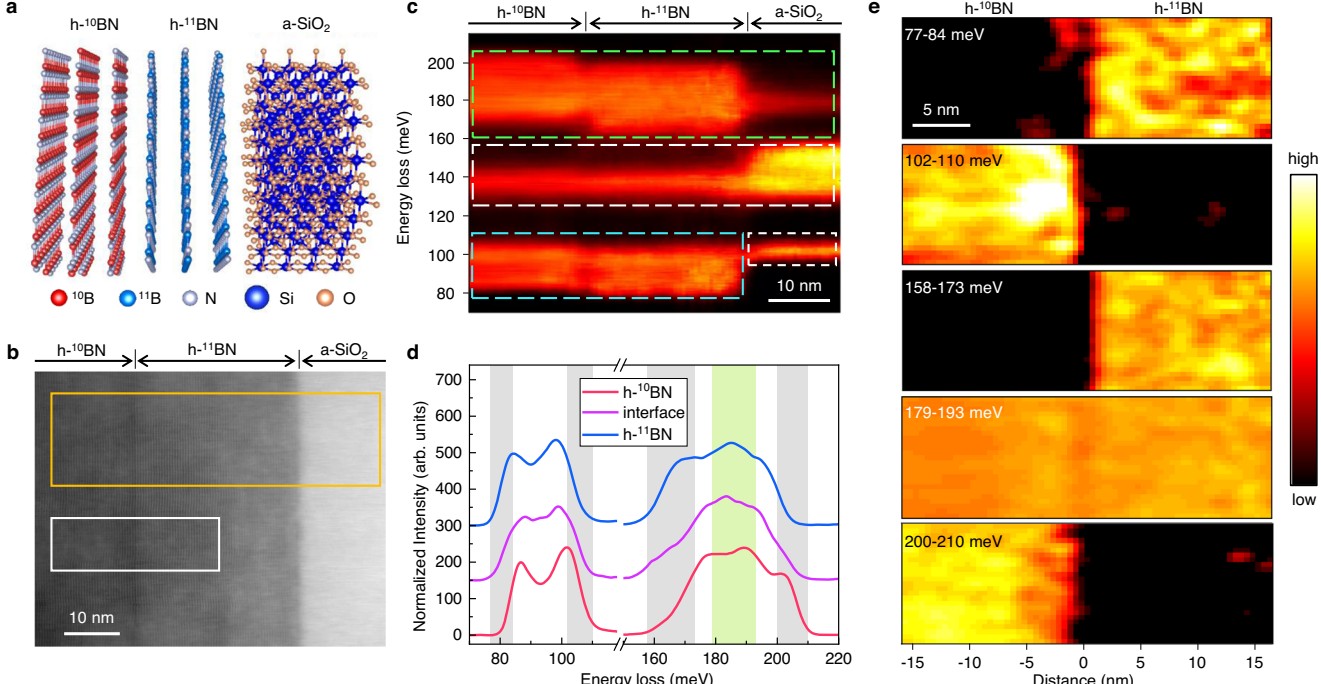

**Fig. 1 | Isotope identification of h−$^{10}$BN/h−$^{11}$BN interface. a** Schematic of the assembled stack of h-$^{10}$BN, h-$^{11}$BN, and Si substrate with amorphous SiO$_2$ (a-SiO$_2$) layer on surface. **b** HAADF image of the h-$^{10}$BN/h-$^{11}$BN/a-SiO$_2$ cross-section. **c** STEM-EELS vibrational spectra were acquired at the region of the orange rectangle labeled in **b**. The vibrational signals from the h-BN in-plane direction, a-SiO$_2$ and h-BN out-of-plane direction are labeled by green, white and blue dashed rectangles, separately. **d** STEM-EELS vibrational spectra. The red, purple, and blue solid lines are acquired from h-$^{10}$BN, interface, and h-$^{11}$BN, separately. The details of acquisitions and processing procedures are included in methods. **e** Energy-filtered images of h-$^{10}$BN/h-$^{11}$BN. The acquisition region is labeled by the white rectangle in **b**, and the energy selection windows are shown as vertical shades in **d** and labeled on the left-up corner of each map.

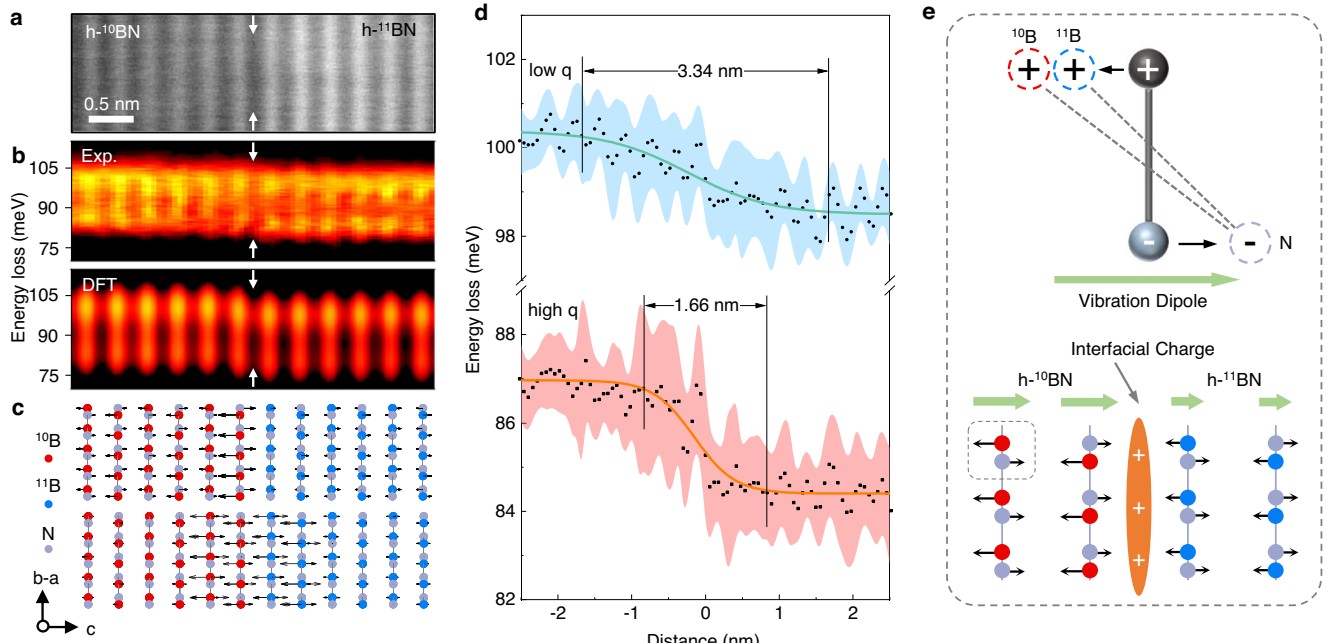

**Fig. 2 | Atomically quantitative analysis of ZO modes at h-$^{10}$BN/h-$^{11}$BN interface.**
**a** HAADF image of h-$^{10}$BN/h-$^{11}$BN interface. **b** The corresponding experimental EELS obtained at the same region of **a**, and the corresponding first-principles calculation results with the same experimental parameters. See analysis details in methods.
**c** Spatial distribution of typical phonon modes around the h-$^{10}$BN/h-$^{11}$BN interface. The arrows are eigenvectors extracted from DFT calculations corresponding to the Brillouin Zone center (q = Γ, ω = 102.5 meV) and Brillouin Zone boundary (q = K, ω = 71.2 meV & ω = 74.7 meV), denoting the two out-of-plane vibrations in **b**.
**d** Quantitative energy variation of the ZO modes. The black dots are fitted phonon

energy of each spectrum, and the error bars are standard deviation shown as the red and cyan shades. The orange and cyan solid lines are fitting of black dots by the Logistic function, presenting the transition width of ZO$_{low q}$ (Brillouin Zone center) and ZO$_{high q}$ (Brillion Zone boundary) is 3.44 nm and 1.66 nm respectively. See analysis details in methods. **e** Schematic of phonon-induced vibration dipole and accumulated bound charge at interface caused by the discontinuity of atom displacement. Detailed spatial distribution of the charge density refer to supplementary fig. 9.

for the in-plane phonon peaks and ~4 meV for the out-of-plane phonon peaks between these two isotopes, which should be sufficiently large to be captured by the STEM-EELS[40,43].

Figure 1c shows the vibrational EELS acquired across the heterostructure (orange rectangle in Fig. 1b). The vibrational signals from the in-plane modes of h-BN (160-200 meV), a-SiO$_2$ (125-155 meV, 100 meV), and out-of-plane modes of h-BN (80-100 meV) are outlined by green, white and blue dashed rectangles, separately. Note that the phonon polaritons of h-BN (~180 meV)[44] and a-SiO$_2$ (~130 meV) are also observable. For the analysis below, we carefully exclude the phonon polariton signals and focus on the h-BN phonons (see details in supplementary fig. 3 and supplementary note 2). Figure 1d shows typical vibrational spectra at h-$^{10}$BN, h-$^{11}$BN regions and interface, denoted by the red, blue, and purple solid lines, respectively. For h-$^{10}$BN, there is a blue shift of ~6.1 meV for the in-plane band and ~3.3 meV for the out-of-plane band compared to that of h-$^{11}$BN. The interfacial spectrum basically lies between the two bulk spectra. EELS mappings with different energy windows (marked by the colored shades in Fig. 1d) are displayed in Fig. 1e, corresponding to the acquisition region denoted by the white box region in Fig. 1b. The interface can be distinguished owing to the different vibration energies between these two isotopes. Note that the h-BN phonon polariton signals at 179–193 meV present the homogenous intensity over the entire field of view due to the highly delocalized nature of phonon polaritons (hundreds of nanometers). It also should be pointed out that during the sample transfer process, a twist between two h-BN isotopic nanosheets is introduced, for which we have experimentally measured (here ~10°) and compared with the DFPT calculation to evaluate the possible effect on the quantitative analysis (see details in supplementary fig. 4 and supplementary note 3). We find that although the in-plane phonons are sensitive to the phonon polaritons of h-BN[44], as shown in Fig. 1c, d and

supplementary fig. 4, the out-of-plane modes are not. Nevertheless, our experimental data proved the influence of tilt angle to phonon energy is neglectable (see supplementary fig. 5). Below we quantitatively analyze ZO phonon changes across the isotopic interface.

**Phonon transition across the isotope interface**

Figure 2 shows the atomically resolved EELS of ZO phonons across the isotope interface. Figure 2a is the HAADF image of the spectrum region and Fig. 2b is the line profile of ZO phonons across the interface with the corresponding simulation data shown below (details in method). The white arrows highlight the isotope interface position. Two main spectral features, whose energy lies between 95-105 meV and 80-90 meV, originate from different momentum transfer, i.e., q near Γ (BZ center, low q) and q near K (BZ boundary, high q) by comparing the out-of-plane signals of DFPT results (supplementary fig. 2). Thus, the two optical phonon modes are labeled as ZO$_{low q}$ and ZO$_{high q}$ respectively. Figure 2c shows two typical eigenvectors of these two ZO modes respectively (modes selection shown in supplementary fig. 6). In Fig. 2b, both ZO$_{low q}$ and ZO$_{high q}$ spectra exhibit a substantial variance of intensity with the atomic period, in good agreement with the DFPT results, indicating the experimental EELS results have achieved a sub-unit-cell scale (<0.34 nm) in probing the isotope signals.

Interestingly, across the interface the transitions of both these ZO phonons are not atomically sharp in Fig. 2d, and their transition widths are different, i.e., ~3.34 nm for ZO$_{low q}$ and ~1.66 nm for ZO$_{high q}$, both of which exceed a single atomic layer ~0.33 nm for h-BN (details of transition width fitting are shown in methods). In contrast, the transition length for LO/TO phonons are only ~0.67 nm and ~0.51 nm, respectively (see supplementary fig. 7). The ground state DFT calculations in supplementary fig. 8 suggested that the transition should be atomically sharp, suggesting the involvement of effects beyond harmonic

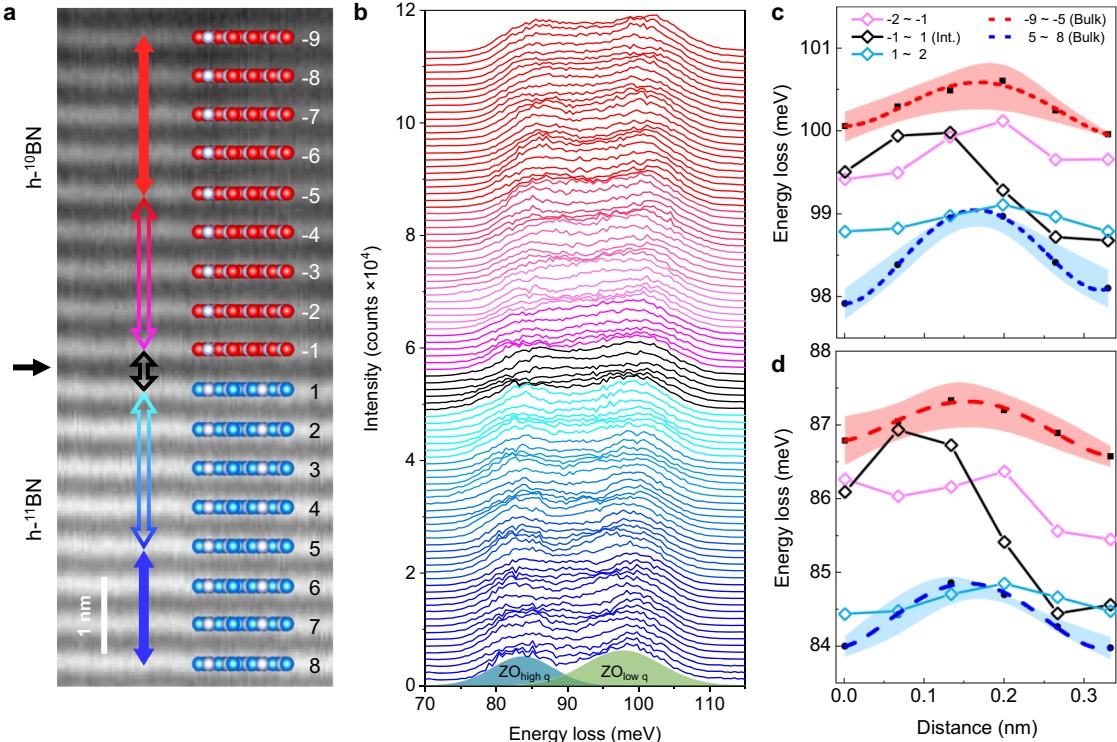

**Fig. 3 | Vibration energy change between atomic layers. a** HAADF image and the schematic of the structure of the h-$^{10}$BN/h-$^{11}$BN interface. Each atomic layer is labeled as the number of layer away from interface (plus for h-$^{11}$BN and minus for h-$^{10}$BN). The interface is between −1 and 1 layer. **b** EEL spectra of ZO phonons from the h-$^{10}$BN to h-$^{11}$BN. The acquision position of spectra are labeled in **a** by the arrows with the same color, i.e. the spectra of interface (layer No. −1 to 1) are shown in black, the adjacent layers to the interface (layer No. −2 to −1 and No. 1 to 2) are shown in magenta and cyan, and the bulk h-$^{10}$BN (layer No. −9 to −5) and h-$^{11}$BN (layer No. 5 to 8) are shown in red and blue. No. "n to n + 1" lable the spatra that are

acquired between n and n + 1 atomic layers. The bottom shows the fitted ZO$_{high\ q}$ and ZO$_{low\ q}$ modes by the multi-gaussian method. **c, d** The vibration energy change of **c** ZO$_{low\ q}$ and **d** ZO$_{high\ q}$ between the atomic layers. The starting position 0 corresponds to the No. n atomic layer, and the ending position coreponds to the No. n + 1 layer. The black dots are averaged phonon energy of multi layers in h-$^{10}$BN and h-$^{11}$BN, the red and blue dashed lines are fitted curves. The orange and cyan bands are the error bars calculated by standard deviation. Black (layer No. −1 to 1), magenta (layer No. −2 to −1) and cyan (layer No. 1 to 2) solid lines are data at the interface and two adjacent layers.

approximation and Born-Oppenheimer approximation harmonic phonon on the electronic ground state.

To explore the electron–phonon coupling at the interface, we calculated the phonon-induced differential charge density at finite temperature. When considering vibration of optical modes in polar material h-BN, the positively charged B and negatively charged N atoms are displaced from the equilibrium position with different amplitudes and opposite directions, thus inducing an additional dipole (vibration dipole)[26], as illustrated in Fig. 2e. At the interface of isotope heterostructure, the change of mass causes a discontinuity of vibration amplitude[45], thus further causes a gradient of vibration dipole as well as accumulation of bound charge at the interface (Fig. 2e). To evaluate this effect, we calculated the differential charge density induced by each ZO mode under 300 K (see supplementary note 1). The line profile of differential charge density induced by ZO modes at q = Γ, q = M and q = K are shown in supplementary fig. 9, respectively. As can be seen, interfacial charge induced by ZO phonon at BZ center is higher than that at BZ boundary, indicating stronger interfacial electron-phonon coupling of ZO$_{low\ q}$. Such a phonon-induced differential charge density at the isotope interface is another isotope effect due to the electron-phonon interaction, which in turn affects the lattice vibration at the interface.

As a comparison, we performed a similar experiment on a regular h-BN/graphite heterostructure, and found that the transition lengths for ZO phonons is only about two atom layers, much more localized than the case in h-$^{10}$BN/h-$^{11}$BN. This is likely due to the C atoms in graphite are neutral so the phonon-induced dipole in h-BN can hardly affect the vibration in graphite (see supplementary fig. 10).

## Sub-unit-cell analysis of phonon energy at the interface

We further explore the changes in phonon energy between atomic layers near the interface. Figure 3a shows a HAADF image of the h-$^{10}$BN/h-$^{11}$BN interface, wherein each atomic layer is labeled. In Fig. 3b, the phonon energy line profiles plotted from h-$^{10}$BN to h-$^{11}$BN show the phonon energy changes between the adjacent atomic layers. Specifically, from the "on-column" position to the "off-column" position, the measured phonon frequency is slightly increased. The quantitative analysis shown in Fig. 3c, d (see details in supplementary fig. 11), where the averaged spectra from the bulk region with shadowed standard deviation error band, show that for both ZO$_{low\ q}$ and ZO$_{high\ q}$ the vibration energy goes up first and then goes down from one atomic layer to another. There is an energy difference -0.68 meV of ZO$_{low\ q}$ between the on-column and off-column cases, which can be interpreted by the change in effective scattering angle[35]. However, the variation in ZO$_{high\ q}$ (Fig. 3d) between atomic layers is not as significant as that in ZO$_{low\ q}$ (Fig. 3c) because the signal of ZO$_{high\ q}$ comes from electrons scattered from the BZ boundary, where the phonon dispersion is flat so that the change of momentum transfer affects the energy little.

The phonons at the interface (black lines in Fig. 3c, d) are affected by both atomic scattering and mass change. Thus the vibration energy shows a combined and complicated manner. With the difference in the distance from the atomic interface, the dominant effect of scattering changes. As the probe moves from h-$^{10}$BN to h-$^{11}$BN, the cross-section of the electron is dominant when near the atomic layer, so the vibration energy rises first. Gradually approaching the h-$^{11}$BN layer, the influence of atomic mass change becomes dominant, thus the energy

gradually decreases. The energy of the adjacent layers near the interface is also affected, consistent with results in Fig. 2.

## Discussion

For an isotope-purified h-$^{10}$BN/h-$^{11}$BN interface system, the lattice vibrations near the interface are probed with sub-unit-cell resolution. Meanwhile, the phonon energies of ZO$_{high\ q}$ and ZO$_{low\ q}$ are different, allowing us to reveal the momentum transfer-dependent behavior as well. We find that the ZO phonon modes at the interface are delocalized to a few nanometers, and the BZ center's phonons are more delocalized than those at the BZ boundary. We also find the electron-phonon coupling at the BZ center is stronger than that at BZ boundary. Moreover, the vibration energy of ZO phonons also varies from one atomic column to another, i.e., it reaches the maximum in the middle of them, which can be understood by the varying effective scattering angle during electron beam scattering. Such behavior also depends on the momentum transfer, that is, at the BZ boundary the flatter phonon dispersion makes the vibration energy less sensitive to the change of scattering angle in different positions. Across the interface from the h-$^{10}$BN layer to h-$^{11}$BN layer, the vibrational energy of two ZO phonons dramatically changes because both of the simultaneous changes in effective scattering angle and mass across the interface.

It is generally believed that the isotopes have little influence on electronic structures. A previous study reported that the optical band gap energies of h-BN depend on the isotopic composition[2], suggesting a strong nuclei effect on the electronic structures. Our study indicate that the electronic structure can be also affected by the isotopes but in a different manner, i.e., isotopes have different vibration amplitude and thus different vibration dipole, leading to the charge accumulation at their interface due to the discontinued van der Waals bonding, which in turn, delocalize the vibration behavior. More interestingly, such behavior is momentum-transfer dependent. Besides the phonon transport, the accumulated charge at the isotope interface likely also influences on the material electron activities.

On the other hand, it's well known that the isotope components govern the thermal conductivity of a material, and the well-accepted mechanism is that the mass disorder dampened the phonon transport which subsequently decreases the thermal conductivity[24,25]. For an ideal interface, a native model would assume an atomically abrupt change of phonon energy which leads to the reduction of phonon transport channels due to reduced overlap of phonon DOS based on the diffuse mismatch model[46,47]. However, the delocalization of lattice vibration at the isotope interface revealed in our work, is similar to that of the localized interface phonon modes for regular heterostructures such as c-BN/diamond[48] and Si/Ge[49], which can act as phonon bridges[50] to promote heat transport, suggesting isotope interface as a new degree of freedom for engineering of heat transport. On the other hand, in thermoelectrics, the highly desired suppression of delocalization phonons is generally realized by the introduction of structural defects and alloying-induced disorder. In this sense, our study suggests that the elimination of the isotope interface is also essential. Moreover, considering the phonon group velocity also highly relies on the momentum transfer, the momentum transfer dependent delocalization of phonons at the isotope interface will affect heat transport in an even more complicated manner.

In summary, by using the advanced vibrational STEM-EELS, we simultaneously achieve the sub-unit-cell spatial resolution and momentum resolution for the phonon measurement across an isotopic interface h-$^{10}$BN/h-$^{11}$BN. We find that the out-of-plane phonons at the isotopic interface are significantly delocalized compared to in-plane ones and phonons at the regular heterointerface. Moreover, for the out-of-plane phonons at Brillouin zone center have longer transition length than that at the Brillouin zone boundary. These phenomena can be understood by the electron-phonon coupling at the isotopic

interface. These findings provide a new angle rather than a simple mass disorder to understand the isotopic effects on the physical properties in natural materials, and further shed light on tailoring material properties via proper engineering of isotopic interfaces.

## Methods

### Crystal growth of isotopically tuned h-BN and TEM sample preparation

The enriched h-BN samples were synthesized using Fe flux method[26,51]. High purity $^{10}$B (97.18 at %, 3 M) or $^{11}$B (99.69 at %, 3 M) and Fe (99.9%, Alfa Aesar) powders were mechanically mixed, and then loaded into an alumina crucible and placed at the center of a high-temperature single-zone tube furnace. The furnace was evacuated and then filled with N$_2$ (5 % H$_2$ mixed) and argon to atmospheric pressure. The furnace was heated to 1550 °C for a dwell time of 24–48 h, ensuring precursors and Fe flux formed a complete solution. After that, a slow cooling to 1450 °C with a rate of 4 °C/hour is applied to obtain high-quality crystals. The furnace is then quenched (300 °C/hour) to room temperature. During the growth process, the N$_2$ (5 % H$_2$ mixed) and argon continuously flowed through the system at rates of 95 sccm and 5 sccm, respectively. The isotopically mixed h-BN crystals as required can be obtained by simply customizing the ratio of $^{10}$B and $^{11}$B powders.

The van der Waals heterostructures of h-$^{10}$BN/h-$^{11}$BN were prepared with polycarbonate (PC) thin film by a pick-up method. The h-BN nanosheets were first exfoliated on silicon substrates with a 300 nm-thick oxide layer. We used PC thin film to pick up the top h-$^{10}$BN nanosheet. The PC thin film with the h-$^{10}$BN nanosheet was then stamped onto the bottom h-$^{11}$BN nanosheet with accurate alignment control based on a homemade transfer stage and optical microscope. Then we dissolved the PC in trichloromethane for 12 h at room temperature. Finally, the as-transferred heterostructure was thermally annealed at 400 °C for 2 h under a high vacuum to further clean the surface. The stacked h-$^{10}$BN/h-$^{11}$BN nanosheet was then made into a TEM cross-section sample by focused ion beam (FIB). The transfer procedure for h-BN/graphite sample and TEM sample preparation procedures are the same as for h-$^{10}$BN/h-$^{11}$BN.

### EELS and imaging experiments

The EELS experiments were carried out on a Nion U-HERMES200 electron microscope with a monochromator that operated at 30 kV and 60 kV to avoid damaging BN materials. For the sub-unit-cell resolution STEM-EELS experiments, we employed a convergence semi-angle $\alpha$ = 35 mrad and a collection semi-angle $\beta$ = 25 mrad, operated at 60 kV. In this setting, the spatial resolution was better than 0.1 nm, while the energy resolution was ~8 meV. The beam current used for EELS was ~10 pA, and the acquisition times were 200–600 ms/pixel.

### EELS data processing

All the acquired vibrational spectra were processed by using the custom-written MATLAB code and Gatan Microscopy Suite. More specifically, the EEL spectra were firstly aligned and then normalized to the intensity of the ZLP. Subsequently, the block-matching and 3D filtering (BM3D) algorithms were applied for removing the Gaussian noise. The background arising from both the tail of the ZLP and the non-characteristic phonon losses was fitted by employing the modified Pearson-VII function with two fitting windows and then subtracted in order to obtain the vibrational signal[48]. The Lucy-Richardson algorithm was then employed to ameliorate the broadening effect induced by the finite energy resolution, taking the elastic ZLP as the point spread function. The spectra were summed along the direction parallel to the interface for obtaining line-scan data with a good signal-to-noise ratio. In addition, we employed a multi-Gaussian peak fitting method to extract the polariton peaks from the composed signal.

## Interface transition width fitting

The transition width of out-of-plane phonon modes is evaluated by a Logistic function, which is widely used to track the transition between two states. The formula of the employed fitting function is

$$y = \frac{a}{e^{\frac{(x-b)}{c}} + 1} + d \tag{1}$$

where $a, b, c, d$ are the fitting parameters. Among the four fitting parameters, $a$ and $d$ are used to normalize the dataset, $b$ is the center of two states, and $c$ is related to the conversion speed from one state to another. When the bias to the center is larger than $|\pm 3c|$ then the concentration of one state is larger than 97.6%, thus we use $6c$ as the transition width.

## Data availability

The EELS data generated in this study have been deposited in the Open Science Framework database under accession code https://osf.io/zwxnu/. The other data that support the findings of this study are available from the corresponding author upon request.

## Code availability

A GUI version of the MATLAB code for the EELS data processing can be found on GitHub at https://github.com/ruishiqi/EELS. Other custom MATLAB codes that is used for DFT related post-processing are available from the corresponding author upon request.

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

## Acknowledgements

This work was supported by the National Key R&D Program of China (2021YFA1400500 to J.C., L.L., and E.G.W.), the National Natural Science Foundation of China (52125307 to P.G., 11974023 to P.G., 52021006 to P.G., 11974024 to J.C., 92165101 to J.C., 52025023 to K.L., U1932153 to L.L., 11974001 to L.L., and T2188101 to P.G.), Guangdong Major Project of Basic and Applied Basic Research (2021B0301030002) to K.L., the Strategic Priority Research Program of Chinese Academy of Sciences under Grant No. XDB33000000 to J.C. and E.G.W., and the "2011 Program" from the Peking-Tsinghua-IOP Collaborative Innovation Center of Quantum Matter. We acknowledge the Electron Microscopy Laboratory at Peking University for using electron microscopes, and the High-performance Computing Platform of Peking University for providing computational resources.

## Author contributions

P.G., L.L., J.C., and E.G.W. conceived the idea; EELS experiments were performed by N.L. and R.S. under the direction of P.G.; growth of pure h-BN isotopes and the fabrication of isotopic h-BN and h-BN/graphite heterostructure were performed by Y.L. under the direction of L.L.; TEM samples were prepared by Z.L. and F.L.; DFT calculations were performed by R.S. and R.Q. under the direction of J.C. and E.G.W.; data processing and analysis were performed by N.L. and R.S., with assistant from R.Q., Y.L., X.G., and X.Z.; N.L. and R.S. initially wrote the manuscript with input from P.G., L.L., Y.L., K.L., Y.J., and X.Z.L.; The revisions were done by R.S. with the assistance of N.L. and Y.L. under direction of J.C., L.L., and P.G.; N.L., R.S. and Y.L. contributed equally to this work. E.G.W. supervised the project.

## Competing interests

The authors declare no competing interests.
