## [Peer Review File · Nature Communications]

Phonon transition across an isotopic interfaceREVIEWER COMMENTS

Reviewer #1 (Remarks to the Author):

A manuscript by Li et al reported a work about the EELS study of isotope labeled hBN heterostructures. Particularly, the authors studied the vibration spectra at the h-10BN/h-11BN interface and found strong delocalization of the out-of-plane optical phonons. By using the unique characterization tool and first-principles calculations, they interpreted the data and especially proposed the possible influence of the isotope upon the electronic structures and thermal conductivity. This work may have fundamental importance to the isotopic effects in the field of solid state physics and the experimental methodology is novel and interesting. I therefore recommend that the work can be published in Nature Communications after addressing the following issues and comments.

1. Throughout the context, the authors may double check the words and grammar. For example, "nucleation process" in the abstract part, may confuse the readers about the research field and directions. In my opinion, "nucleation" is a process during the crystal growth.
2. In figure 1c, the colors of three dashed rectangles are not clear, the authors should improve it.

Reviewer #2 (Remarks to the Author):

This work presented an interesting study of lattice vibration behavior at the isotopic interface of h10BN/h11BN heterostructure. They found the out-of-plane optical phonons at the interface have a wide transition regime, with phonons near the Brillouin zone center have a larger transition regime compared to the phonons at the Brillouin zone boundary. They proposed that the isotope-induced charge effect at the interface accounted for the distinct behavior. From the theoretical aspect, I have the following comments:

1. As the authors said, for regular heterostructures the interface phonon modes are often localized while for h10BN/h11BN heterostructure, the ZO modes are kind of delocalized. The disagreements between the ground state DFT calculations and experimental measurements suggested the involvement of effects beyond harmonic approximation and Born-Oppenheimer approximation, i.e., electron-phonon coupling effects. The authors further proposed that isotopes have different vibration amplitude and thus different vibration dipole, leading to the charge accumulation at their interface due to the discontinued van der Waals bonding, which in turn, delocalize the vibration behavior. Such explanations are not convincing because similar things also happen for regular heterostructures, i.e., MoS2/WS2 and BN/graphene, which would have much larger charge accumulation. I hope the authors can clarify the essential difference.
2. According to the Frohlich interaction, the coupling between electrons and longitudinal optical phonons is proportional to $1/q$, where q is the momentum of phonons. It seems that the stronger the electron-phonon coupling, the more delocalized of the ZO modes according to the statements of the authors. This point is very strange. Why ZO modes are special and why not other modes? And again, why ZO modes in h10BN/h11BN but not in other regular heterostructures?
3. Another strange point is that the authors said the twist between h10BN and h11BN will not affect the out-of-plane modes. As we know, twist will change the electronic properties and thus electron-phonon couplings. In this sense, twist should affect out-of-plane modes as well and DFPT calculations might be not convincing as no electron-phonon coupling is considered. Is there any experimental evidence?
4. Some other comments are: the title of the main manuscript is not the same as that of the supplemental materials; in Fig. 1c, the colors of the lines are very difficult to recognize.

In general, I think this work presents an interesting experimental phenomenon about the isotope effects at the interface but a clearer understanding is still lacking at this stage.

Reviewer #3 (Remarks to the Author):

The manuscript entitled "Phonon transition across an isotropic interface" investigate the phonon transition at h-10BN/h-11BN interface using electron energy loss spectroscopy in scanning transmission electron microscopy. The authors emphasized the importance of the isotope through exotic phenomena such as different superconducting transition temperatures and substantial thermal conductivity reduction. However, this manuscript lacks significantly in terms of include those exotic phenomena. The authors focus on describing the phonon transition at the hBN isotope interface by different magnitude EELS-STEM measurements and their corresponding DFT calculations. This topic only has a limited audience. Overall, I think this manuscript does not have enough novelty for publication in Nature Communications.

I leave some comments to improve this manuscript.

1. The authors emphasize the importance of isotope but did not explain the importance of isotope interface. Thus, the results not correlated with the introduction part.
2. Why there is non-uniform charge density fluctuation in each h10BN and h11BN side (the regions far-away from the interface) in Fig. S7?
3. In Fig. 2, the authors mentioned, "the transition width is ~ 3.34 nm for ZO low q ". However, the regions 1-2 and 5-6 in Fig. 3 include this range. Therefore, the author should provide far-away regions as bulk, instead of 1-2 and 5-6 regions.
4. As a minor comment, the figures quality should be improved to show their results more clearly. For example, 1) Fig.1c the rectangular box is not clear. 2) Fig.1d EELS spectra should be stacked at some distance, and so on.

Response to reviewers' comments

We would like to thank three reviewers for their thorough and highly constructive comments concerning our manuscript. In response to the comments and suggestions, we have provided new experimental and theoretical data, conducted further analysis, and thus enhanced our understanding and improved the manuscript. We believe that all the concerns raised by the referees are addressed appropriately in the revised manuscript.

Major points and key issues:

The reviewers mainly raised questions on our explanation and understanding of delocalized phonon transport at the isotope interface, and the novelty of our study. We have provided new experimental and theoretical data to enhance the reliability of our explanation, i.e., the comparison between phonon behavior in isotope heterostructure interface and regular heterostructure h-BN/graphite interface, and the different behavior between in-plane modes and out-of-plane modes in h-¹⁰BN/h-¹¹BN. In the revised manuscript, we added these data as two supplemental figures. We have also modified our introduction and discussion thus made better clarifications of our novelty. Besides, we have also added details in the results, discussion, and method sections.

Major changes in the revision:

According to the referees' suggestions and comments, we have revised and improved the manuscript. Changes are highlighted with blue font color in the revised manuscript. Below is a list for major ones.

1. To clarify the difference between an isotope heterostructure interface and regular heterostructure interface, we performed a similar STEM-EELS experiment at a h-BN/graphite interface, and found the phonon transport are quite localized, which is in favor of our explanation that phonon delocalization is caused by

electron-phonon coupling. We have added this result as supplementary figure 10 in the revised manuscript, and add some discussion comparing between regular heterostructure and isotope heterostructure in the manuscript.

2. We made similar analysis to the energy changes of LO/TO modes from the same dataset that is analyzed in manuscript. The transition length for LO/TO phonons are ~ 0.67 nm and ~ 0.51 nm, respectively, much shorter than ZO modes. This agrees our explanation. We have added this result as supplemental figure 7 in the revised manuscript.
3. We have corrected the words causing the confusion of the tilt angle between electron beam and sample, and the twist angle between two h-BN flakes. We also made further experiments to prove the neglectable influence of tilt angle to our conclusion, and added this result as supplemental figure 5 in the revised manuscript.
4. We have modified our introduction and discussion thus made better clarifications of our novelty.
5. According to the reviewers' suggestions, we showed the full EEL spectra including the regions that is far-away from the interface, and replace the bulk spectra of h- ^{10}BN and h- ^{11}BN in Fig. 3c-d with the averaged spectra of far-away from the interface.
6. According to the reviewers' suggestions, we changed the colors and border width of dashed rectangles in figure 1c, stacked EELS spectra in figure 1d with a vertical offset, and improved the quality of figures.
7. Other changes following referee's suggestions. Please see our response below for detail.

Point-by-point response:

Response to reviewer 1

A manuscript by Li et al reported a work about the EELS study of isotope labeled hBN heterostructures. Particularly, the authors studied the vibration spectra at the h-¹⁰BN/h-¹¹BN interface and found strong delocalization of the out-of-plane optical phonons. By using the unique characterization tool and first-principles calculations, they interpreted the data and especially proposed the possible influence of the isotope upon the electronic structures and thermal conductivity. This work may have fundamental importance to the isotopic effects in the field of solid state physics and the experimental methodology is novel and interesting. I therefore recommend that the work can be published in Nature Communications after addressing the following issues and comments.

Response: We would like to thank the reviewer for the recommendation for the publication and positive comments on the importance of our work.

1. Throughout the context, the authors may double check the words and grammar. For example, “nucleation process” in the abstract part, may confuse the readers about the research field and directions. In my opinion, “nucleation” is a process during the crystal growth.

Response 1: We thank the reviewer for the careful reading and kind suggestion. We have changed “*nucleation process*” to “*nuclear process*” to correct the typo error in the manuscript. We have also polished the language thoroughly in the revised manuscript.

2. In figure 1c, the colors of three dashed rectangles are not clear, the authors should improve it.

Response 2: We greatly appreciate the reviewer’s valuable suggestion. We have made adjustments according to the reviewers’ suggestions, changed the colors and border width of dashed rectangles in figure 1c, and improved the quality of figures, which should make the figure clearer now. The modified figure 1 is shown below.

Figure 1 Isotope identification of h-¹⁰BN/h-¹¹BN interface. (a) Schematic of the assembled stack of h-¹⁰BN, h-¹¹BN and Si substrate with amorphous SiO₂ (a-SiO₂) layer on surface. (b) HAADF image of the h-¹⁰BN/h-¹¹BN/a-SiO₂ cross-section. (c) STEM-EELS vibrational spectra were acquired at the region of the orange rectangle labeled in (b). The vibrational signals from the h-BN in-plane direction, a-SiO₂ and h-BN out-of-plane direction are labeled by green, white and blue dashed rectangles, separately. (d) STEM-EELS vibrational spectra. The orange, green, and purple solid lines are acquired from h-¹⁰BN, interface, and h-¹¹BN, separately. The details of acquisitions and processing procedures are included in methods. (e) Energy-filtered imaging of h-¹⁰BN/h-¹¹BN. The acquisition region is labeled by the white rectangle in (b), and the energy selection windows are shown as vertical stripes in (d) and labeled on the left-up corner of each map.

Response to reviewer 2

This work presented an interesting study of lattice vibration behavior at the isotopic interface of $h\text{-}^{10}\text{BN}/h\text{-}^{11}\text{BN}$ heterostructure. They found the out-of-plane optical phonons at the interface have a wide transition regime, with phonons near the Brillouin zone center have a larger transition regime compared to the phonons at the Brillouin zone boundary. They proposed that the isotope-induced charge effect at the interface accounted for the distinct behavior. From the theoretical aspect, I have the following comments:

1. As the authors said, for regular heterostructures the interface phonon modes are often localized while for $h\text{-}^{10}\text{BN}/h\text{-}^{11}\text{BN}$ heterostructure, the ZO modes are kind of delocalized. The disagreements between the ground state DFT calculations and experimental measurements suggested the involvement of effects beyond harmonic approximation and Born-Oppenheimer approximation, i.e., electron-phonon coupling effects. The authors further proposed that isotopes have different vibration amplitude and thus different vibration dipole, leading to the charge accumulation at their interface due to the discontinued van der Waals bonding, which in turn, delocalize the vibration behavior. Such explanations are not convincing because similar things also happen for regular heterostructures, i.e., MoS_2/WS_2 and $\text{BN}/\text{graphene}$, which would have much larger charge accumulation. I hope the authors can clarify the essential difference.

Response 1: We really appreciate the reviewer's comment on this. It would be greatly helpful in studying the interface phonon transport at other regular heterostructure interfaces as a comparison. However, the transition length in our result is within several nanometers, which is not detectable for other experimental techniques. Therefore, we carried out a similar STEM-EELS experiment at a $h\text{-BN}/\text{graphite}$ interface. The HAADF image of the $h\text{-BN}/\text{graphite}$ interface is shown in Figure R1a, where the EELS acquisition region is labeled as red dashed rectangle. Like $h\text{-BN}$, the ZO phonons of graphite can also be classified to $\text{ZO}_{\text{low } q}$ and $\text{ZO}_{\text{high } q}$, as shown in Figure R1b. The fitted $\text{ZO}_{\text{low } q}$ and $\text{ZO}_{\text{high } q}$ phonon energy change across $h\text{-BN}/\text{graphite}$ interface is shown in Figure R1c. We find the transition lengths of these modes are ~ 0.63 nm and ~ 0.75 nm respectively, which is only about 2-3 times of the atomic interlayer distance, much shorter than that in $h\text{-}^{10}\text{BN}/h\text{-}^{11}\text{BN}$ system. Moreover, the transition length for $\text{ZO}_{\text{low } q}$ and $\text{ZO}_{\text{high } q}$ modes are very close, which is very different from the behavior in $h\text{-}^{10}\text{BN}/h\text{-}^{11}\text{BN}$ interface where the transition length differs a lot. This is possibly due to the C atoms in graphite are neutral so the phonon induced dipole in $h\text{BN}$ can hardly affect the vibration in graphite.

Figure R1 | The phonon transport of ZO modes in h-BN/Graphite. **a.** the HAADF image of h-BN/graphite heterostructure. The white dashed line indicates the interface, and the red dashed rectangle labels the EELS acquisition region. **b.** the phonon dispersion of graphite (red) and h-BN (blue), and the projected phonon density of states for graphite $ZO_{low\ q}$ (red dashed line), graphite $ZO_{high\ q}$ (red solid line), h-BN $ZO_{low\ q}$ (blue dashed line) and h-BN $ZO_{high\ q}$ (blue solid line). **c.** Quantitative energy variation of the ZO modes in h-BN/graphite heterostructure. The black dots are fitted phonon energy of each spectrum, and the error bars are standard deviation shown as the orange and cyan shades. The cyan and orange solid lines are fitting of black dots by the Logistic function, presenting the transition width of $ZO_{low\ q}$ and $ZO_{high\ q}$ is ~ 0.63 nm and ~ 0.75 nm respectively. This is possibly due to the C atoms in graphite are neutral so the phonon induced dipole in hBN can hardly affect the vibration in graphite.

As the reviewer mentioned, our explanations of charge accumulation at interface do occur at regular heterostructures. However, at heterostructure interfaces, the misalignment of fermi level will cause a static charge transfer, which is usually much larger than the dynamical charge accumulation induced by phonon discussed in our manuscript. The most important difference is that the static charge transfer is independent on the phonon modes, so the transition length of phonon modes delocalized by the static charge transfer, if any, may be insensitive to vibrations modes, which agrees with our experimental results on h-BN/graphite interface. But in $h^{10}BN/h^{11}BN$ heterostructure, $ZO_{low\ q}$ and $ZO_{high\ q}$ modes have very different transition length, indicating the different mechanism than the regular heterostructures. Secondly, another significant difference we noticed that is in a regular heterostructure two materials are different and interfacial vibrational modes usually appear (R. Qi et al., *Nature*, 599, 399-403 (2021), C. Zhe et al., *Nat. Comm.*, 12, 6901(2021)) and overwhelm the effects caused of isotope induced charge accumulation. However, this effect is absent in $h^{10}BN/h^{11}BN$ heterostructure, because the atom types are exactly the same at both sides of interface. Only in isotope systems the electron-phonon coupling dominates the phonon transport, and this is the value of studying isotope interfaces.

Revision -1: According to the suggestions from the reviewer, we have added Figure R1 as supplemental figure 10 in the revised manuscript. We've also add the comparison between regular heterostructure and isotope heterostructure in the manuscript Page 7, Line 188-192.

2. According to the Frohlich interaction, the coupling between electrons and longitudinal optical phonons is proportional to $1/q$, where q is the momentum of phonons. It seems that the stronger the electron-phonon coupling, the more delocalized of the ZO modes according to the statements of the authors. This point is very strange. Why ZO modes are special and why not other modes? And again, why ZO modes in h10BN/h11BN but not in other regular heterostructures?

Response 2: We thank the reviewer for pointing out this. We agree the reviewer's point of stronger electron-phonon coupling at low momentum transfer according to the Frohlich theory, as wrote in discussion "*We also find the electron-phonon coupling at the BZ center is stronger than that at BZ boundary.*" (Page 8 Line 223). Actually, this theory helps to explain our experimental result. In our experiment, the delocalization is characterized by the transition length for phonon energy. Our direct observation showed the transition length for $ZO_{\text{low } q}$ is larger than $ZO_{\text{high } q}$, so we propose that the stronger electron-phonon coupling at small q might be the underlying mechanism. This is in accordance with Frohlich theory.

As for the specialness of ZO modes, we first made similar analysis to the energy changes of other modes, i.e., LO/TO modes from the same dataset that is analyzed in manuscript, as shown in Figure R2. The transition length for LO/TO phonons are ~ 0.67 nm and ~ 0.51 nm, respectively, much shorter than ZO modes. This can be understood from the following facts. The vibration directions of ZO modes are out-of-plane ([001] direction), so the dipole induced by ZO modes are also along [001] direction. As the charge is the divergence of dipole, only vibration along [001] direction will cause charge variation between atomic layers, and thus accumulate charge at interface. The other in-plane vibration modes just induce the intralayer charge variation, and will destruct each other when viewed from [100] zone axis. This means the in-plane modes should be localized at the interface, which agrees with our experimental data.

Figure R2 | The phonon transport of LO and TO modes in $h^{10}\text{BN}/h^{11}\text{BN}$. The black dots are fitted phonon energy of each spectrum, and the blue and green solid lines are fitting of black dots by the Logistic function, presenting the transition width of LO and TO is ~ 0.67 nm and ~ 0.51 nm respectively.

And as discussed in the response 1, the electron-phonon coupling effect of ZO modes in other regular heterostructures might be overwhelmed by other larger effects, i.e. the static charge transfer or the formation of new chemical bonding. So our discussed phenomena will only be discovered in isotope interfaces.

Revision -2: To show the different behavior between in-plane and out-of-plane phonon, we have added Figure R2 as supplemental figure 7 in the revised manuscript. Related description is in manuscript Page 6 Line 166-168.

3. Another strange point is that the authors said the twist between $h^{10}\text{BN}$ and $h^{11}\text{BN}$ will not affect the out-of-plane modes. As we know, twist will change the electronic properties and thus electron-phonon couplings. In this sense, twist should affect out-of-plane modes as well and DFPT calculations might be not convincing as no electron-phonon coupling is considered. Is there any experimental evidence?

Response 3: We thank the reviewer for pointing out the influence of twist angle on the electron-phonon coupling, and we are sorry for the improper words and misleading expression. Actually, we did not discuss the dependence of EELS spectra and phonon transition with twist angle of two h-BN crystals based on DFPT calculation. What we were trying to explain is the dependence of EELS spectra with the tilt angle between

electron beam and sample axis of two h-BN flakes. This can be one issue of electron scattering when performing the fine analysis, and also has been well explained using DFPT. (e.g. R. J. Nicholls et al., *Phy.Rev.B*, 99, 094105(2019), R. Senga et al., *Nature*, 573, 247-250 (2019)).

To avoid misunderstanding, we have revised our manuscript in Page 5 Line 140-145 to be:

“It also should be pointed out that during the sample transfer process, a twist between two h-BN isotopic nanosheets is introduced, for which we have experimentally measured (here $\sim 10^\circ$). This will make the tilt angle between electron beam and one side of sample different from the other. We performed the DFPT calculation to evaluate the possible effect on the quantitative analysis (see details in supplemental figure 4 and Note 3).”

We also changed the title of supplemental Note 3 from “Influences of the twist angle on the vibration signals” to “*Influences of the tilt angle on the vibration signals*”.

Moreover, we acquired the EELS spectra of h-BN at tilt angle of 0° , 0.6° and 5.3° . The spectra of $h^{10}\text{BN}$ at these tilt angles (Figure R3a), the quantitative analysis of their ZO energies (Figure R3b) and LO/TO energies (Figure R3c) are shown below. We find the energy change are within 1 meV in experimental tilt angle range, much lower than the energy change caused by the isotope effect (~ 3 meV). We thus conclude the tilt angle will not affect our result.

Figure R3 | The influence of tilt angle on the EELS spectra of h-BN. **a.** The EELS spectra of $h^{10}\text{BN}$ with tilt angle of 0° (orange), 0.6° (purple) and 5.3° (cyan). The gray shades are the standard deviations. **b.** the energy of ZO high-q (magenta) and ZO low-q (blue) modes with tilt angle. **c.** the energy of LO (gray) and TO (yellow) modes with tilt angle. The energy change are within 1 meV in experimental tilt angle range, much lower than the energy change caused by the isotope effect (~ 3 meV).

Revision -3: To better clarify the influence of tilt angle on experiment result, we have added Figure R3 as supplemental figure 5 in the revised manuscript. Related description is in manuscript Page 6 Line 147-148.

4. Some other comments are: the title of the main manuscript is not the same as that of the supplemental materials; in Fig. 1c, the colors of the lines are very difficult to

recognize.

Response 4: We greatly appreciate the reviewer's valuable suggestion. We have made adjustments according to the reviewers' suggestions, corrected the title of supplemental materials, changed the colors and border width of dashed rectangles in figure 1c, and improved the quality of figures, which should make the figure clearer now. The adjusted figure 1 is shown below.

Figure 1 Isotope identification of h-¹⁰BN/h-¹¹BN interface. (a) Schematic of the assembled stack of h-¹⁰BN, h-¹¹BN and Si substrate with amorphous SiO₂ (a-SiO₂) layer on surface. (b) HAADF image of the h-¹⁰BN/h-¹¹BN/a-SiO₂ cross-section. (c) STEM-EELS vibrational spectra were acquired at the region of the orange rectangle labeled in (b). The vibrational signals from the h-BN in-plane direction, a-SiO₂ and h-BN out-of-plane direction are labeled by green, white and blue dashed rectangles, separately. (d) STEM-EELS vibrational spectra. The orange, green, and purple solid lines are acquired from h-¹¹BN, interface, and h-¹⁰BN, separately. The details of acquisitions and processing procedures are included in methods. (e) Energy-filtered imaging of h-¹⁰BN/h-¹¹BN. The acquisition region is labeled by the white rectangle in (b), and the energy selection windows are shown as vertical stripes in (d) and labeled on the left-up corner of each map.

In general, I think this work presents an interesting experimental phenomenon about the isotope effects at the interface but a clearer understanding is still lacking at this stage.

Response: We appreciate the reviewer for the interest on the phenomenon revealed by our manuscript. By the response listed above and the careful revision of our manuscript, we now have a better understanding and clearer clarifications of our discovery, and we

believe our result are able to reach the criterion of publication in Nature Communications.

Response to reviewer 3

The manuscript entitled “Phonon transition across an isotropic interface” investigate the phonon transition at h-10BN/h-11BN interface using electron energy loss spectroscopy in scanning transmission electron microscopy. The authors emphasized the importance of the isotope through exotic phenomena such as different superconducting transition temperatures and substantial thermal conductivity reduction. However, this manuscript lacks significantly in terms of include those exotic phenomena. The authors focus on describing the phonon transition at the hBN isotope interface by different magnitude EELS-STEM measurements and their corresponding DFT calculations. This topic only has a limited audience. Overall, I think this manuscript does not have enough novelty for publication in Nature Communications. I leave some comments to improve this manuscript.

Response: We thank the reviewer for the comment which helped us to improve the novelty clarification. The main novelty of our work is:

1. Isotope analysis techniques have been widely used in various applications, such as chemical/biological tracing or environmental surveys. However, the isotope characterization at nanoscale or even atomic scale is very challenging, since the spatial resolution of normally used methods like mass spectroscopy or Raman spectroscopy are usually limited to a few hundred nanometers. The recent advances STEM-EELS have demonstrated the capabilities to locally detect the distribution and diffusion of isotopes in chemically bonded regions, which draws broad appealing immediately ($^{13}\text{C}/^{12}\text{C}$ -O bonds with spatial distance ~ 50 nm, in *Science* 2019, 363, 525–528; ^{13}C - ^{13}C / ^{12}C - ^{12}C bonds with lattice resolution in *Nature* 2022, 603, 68-72). These studies have focused on local identification of the carbon isotopes. In our study, we further improve the spatial resolution into sub-unit cell level, and elaborately fit the density of states (DOS) of h-BN ZO phonons, which allows us to distinguish the phonons from the Brillouin zone center (low q) and Brillouin zone boundary (high q), respectively. **Thus, for the first time, we simultaneously achieve the sub-unit-cell spatial resolution and momentum resolution for the phonon measurement across an isotopic interface.** Together with former researches, we believe our finding will arouse the interest on the application of atomically resolved isotope tracing in biology process or chemical reaction.
2. Scientifically, we find that the in-plane and out-of-plane phonons at h- ^{10}BN /h- ^{11}BN interface have very different delocalization behavior, i.e., ZO phonons have a wide transition regime for more than ~ 1.6 nm (~ 5 atomic layers) whereas the LO/TO phonons have sharp transition regime within ~ 0.67 nm (~ 2 atom layers). (see Fig 2d in the manuscript and Fig. R2 in response 2 to reviewer#2). Moreover, for the out-of-plane phonons, the phonons at Brillouin zone center or Brillouin zone boundary have distinct transition length. The phenomena can be understood by the electron-phonon coupling at the isotopic interface, which provides new insights into the isotopic effects on the physical properties of impure materials, e.g., isotope scattering on the thermal transport beyond the simple mass disorder.

To make our clarifications better, we've added some discussion on the novelty of our experimental techniques and isotope interface systems (Page 9 Line 260-267). We believe our revised manuscript will not only find broad appealing for the studies related to tuning optical emission, thermo-electronic coefficients and thermal conductivity by isotope engineering, but also demonstrated cutting-edge measurement technique, i.e., resolving isotopes with sub-unit cell space resolution and different momentum-transfer, provides new opportunities to study and utilize the isotopic effects in material science.

1. The authors emphasize the importance of isotope but did not explain the importance of isotope interface. Thus, the results not correlated with the introduction part.

Response 1: We appreciate the reviewer's comment on the importance of isotope interface. In fact, such homo-element but hetero-isotope $h\text{-}^{10}\text{BN}/h\text{-}^{11}\text{BN}$ interface have never been studied from the microscope perspective to the best of our knowledge. Recent studies showed the excitonic phenomena by isotopically purifying materials on the one side of heterostructure, such as regulating interlayer electron-phonon coupling by hyperfine isotope effects (*Nano Lett.* 2022, 22, 2725–2733) and visualizing negative refraction of phonon polaritons (*Science*, 2023, 379, 555-557). Therefore, $h\text{-}^{10}\text{BN}/h\text{-}^{11}\text{BN}$ heterostructure, which is isotopically purified on both sides, is worthy studying. Secondly, for natural materials consisting of isotopic mixtures, nanoscale/atomic-scale heterogeneity leads to emergence of isotope interfaces, which should affect some of material properties such as the phonon transport and thermal conductivity. But technically, this interface effect is very hard to study as they are mixed together and randomly arranged. Here we fabricate artificially stacked and van der Waals bonded $h\text{-}^{10}\text{BN}/h\text{-}^{11}\text{BN}$ interface, where the van der Waals bonded isotopic interface is ideally sharp to avoid any isotope diffusion. This interface is atomic sharp and well aligned, thus can serve as an ideal platform to study the isotope effect at the interface. Our study reveals the mechanism of how isotope interface affects the phonon transport, which shed light on tailoring material properties via proper engineering of isotopic interfaces. To emphasize the importance of isotope interface rather than isotope, we reduced our emphasis on isotope effect, and add some recent advances on the isotope interface in our introduction part (Page 3 Line 69-72):

“Recently, effects of interface with one side of heterostructure is isotopically purified, is studied. Phenomena like hyperfine isotope effects regulating interlayer electron-phonon coupling in heterostructures and visualizing negative refraction of phonon polaritons is reported.”

2. Why there is non-uniform charge density fluctuation in each $h^{10}\text{BN}$ and $h^{11}\text{BN}$ side (the regions far-away from the interface) in Fig. S7?

Response 2: We appreciate the reviewer's careful reading of our manuscript. In our calculation procedure, we were calculating the charge density difference of displaced structures according to the phonon eigenvectors with static structure. Ideally, inside the bulk $h^{10}\text{BN}$ or $h^{11}\text{BN}$ that is far enough away from the interface, the atoms' phonon

eigenvectors should be the same with that of neighbor layers. However, our supercell contains only 10 UC $h^{10}BN/10$ UC $h^{11}BN$, so the phonon modes will interfere with their periodic images due to the periodic boundary condition in DFT framework. This makes the atoms' vibration amplitude variant even away from the interface, causing the non-uniform charge density fluctuation in each $h^{10}BN$ and $h^{11}BN$ side. To verify this, we calculated the differential charge density in a shorter supercell, i.e. 6 UC $h^{10}BN/6$ UC $h^{11}BN$. The line profiles of differential charge density induced by ZO modes at Γ and K point are shown in Figure R4. As expected, the charge fluctuation in 6 UC structure is larger compared to the result of 10 UC in supplemental figure 9a (the scales of y axis in figures are the same). That's because the shorter supercell size will cause more significant interference. Nevertheless, the charge accumulation at Γ point is larger than at K points, indicating stronger electron-phonon coupling of $ZO_{low\ q}$ phonon. This means the size of supercell will not affect our explanation and conclusion in the manuscript.

Figure R4 | Differential charge density in 6 UC $h^{10}BN/6$ UC $h^{11}BN$ structure. The line profiles of differential charge density induced by ZO phonons at (a) Γ point and (b) K point. The charge fluctuation in bulk h -BN away from the interface is larger compared to the result in 10 UC $h^{10}BN/10$ UC $h^{11}BN$ structure. But still, the charge accumulation at Γ point is larger than at K points, indicating stronger electron-phonon coupling $ZO_{low\ q}$ phonon.

3. In Fig. 2, the authors mentioned, “the transition width is ~ 3.34 nm for $ZO_{low\ q}$ ”.

However, the regions 1-2 and 5-6 in Fig. 3 include this range. Therefore, the author should provide far-away regions as bulk, instead of 1-2 and 5-6 regions.

Response 3: We thank the reviewer for the kind suggestion. We do agree the reviewer's comment on the difference on the presented spatial region between Fig.2 and Fig.3, and realized the insufficiency of our presentation in figure 3. As suggested, we showed the full EEL spectra including the regions that is far-away from the interface, and replace the bulk spectra of $h\text{-}^{10}\text{BN}$ and $h\text{-}^{11}\text{BN}$ in Fig. 3c-d with the averaged spectra of far-away from the interface (layer No. -9 ~ -5 and layer No. 5 ~ 8). The revised Figure 3 is shown below.

Figure 3 Vibration energy change between atomic layers. (a) HAADF image and the schematic of the structure of the $h\text{-}^{10}\text{BN}/h\text{-}^{11}\text{BN}$ interface. Each atomic layer is labeled as the number of layer away from interface (plus for $h\text{-}^{11}\text{BN}$ and minus for $h\text{-}^{10}\text{BN}$). The interface is between -1 and 1 layer. (b) EEL spectra of ZO phonons from the $h\text{-}^{10}\text{BN}$ to $h\text{-}^{11}\text{BN}$. The acquisition position of spectra are labelled in (a) by the arrows of the same color, i.e. the spectra of interface (layer No. -1 ~ 1) is shown in black, the adjacent layer to the interface (layer No. -2 ~ -1 and No.1 ~ 2) is shown in magenta and cyan, and the bulk $h\text{-}^{10}\text{BN}$ (layer No. -9 ~ -5) and $h\text{-}^{11}\text{BN}$ (layer No.5 ~ 8) is shown in red and blue. No. “n ~ n+1” label the spectra acquired between n and n+1 atomic layers. The bottom shows the fitted $ZO_{\text{high } q}$ and $ZO_{\text{low } q}$ modes by the multi-gaussian method. (c, d) The vibration energy change of (c) $ZO_{\text{low } q}$ and (d) $ZO_{\text{high } q}$ between the atomic layers. The starting position 0 corresponds to the No. n atomic layer, and the ending position corresponds to the No. n+1 layer. The black dots are averaged phonon energy of multi layers in $h\text{-}^{10}\text{BN}$ and $h\text{-}^{11}\text{BN}$, the red and blue dashed lines are fitted curves. The orange and cyan bands are the error bars calculated by standard deviation. Black

(layer No. -1 ~ 1), magenta (layer No. -2 ~ -1) and cyan (layer No. 1 ~ 2) solid lines are data at the interface and two adjacent layers.

4. As a minor comment, the figures quality should be improved to show their results more clearly. For example, 1) Fig.1c the rectangular box is not clear. 2) Fig.1d EELS spectra should be stacked at some distance, and so on.

Response 4: We greatly appreciate the reviewer's valuable suggestion. We have made adjustment according to the reviewers' suggestions, changed the colors and border width of dashed rectangles in figure 1c, stacked EELS spectra in figure 1d with a vertical offset, and improved the quality of figures, which should make the figure clearer now. The adjusted figure 1 is shown below.

Figure 1 Isotope identification of h-¹⁰BN/h-¹¹BN interface. (a) Schematic of the assembled stack of h-¹⁰BN, h-¹¹BN and Si substrate with amorphous SiO₂ (a-SiO₂) layer on surface. (b) HAADF image of the h-¹⁰BN/h-¹¹BN/a-SiO₂ cross-section. (c) STEM-EELS vibrational spectra were acquired at the region of the orange rectangle labeled in (b). The vibrational signals from the h-BN in-plane direction, a-SiO₂ and h-BN out-of-plane direction are labeled by green, white and blue dashed rectangles, separately. (d) STEM-EELS vibrational spectra. The orange, green, and purple solid lines are acquired from h-¹¹BN, interface, and h-¹⁰BN, separately. The details of acquisitions and processing procedures are included in methods. (e) Energy-filtered imaging of h-¹⁰BN/h-¹¹BN. The acquisition region is labeled by the white rectangle in (b), and the energy selection windows are shown as vertical stripes in (d) and labeled on the left-up corner of each map.

REVIEWERS' COMMENTS

Reviewer #1 (Remarks to the Author):

The authors have addressed all the questions raised by me and the other two reviewers, and the quality of the paper is improved significantly. I, therefore, recommend publication in Nat. Commun.

Reviewer #2 (Remarks to the Author):

All my concerns are solved and I am glad to recommend the publication of this work now.

Reviewer #3 (Remarks to the Author):

In my previous revision, my main concern was regarding the novelty of the work. However, the authors have addressed my questions and comments in detail, and as a result, I now fully understand the importance of their study. Consequently, I believe that this manuscript has the potential to be published in Nature Communications, with only minor revisions.

I leave only one comment to improve this manuscript.

1. To improve readability, would it be possible to make only single energy loss profile along the layers (from -9 to 8) in Figures 3c and d?

Response to referees' comments

We would like to thank three reviewers for taking the time and effort necessary to review our manuscript, and we appreciate their positive feedback.

Point to point responses:

Response to reviewer 3

In my previous revision, my main concern was regarding the novelty of the work. However, the authors have addressed my questions and comments in detail, and as a result, I now fully understand the importance of their study. Consequently, I believe that this manuscript has the potential to be published in Nature Communications, with only minor revisions.

I leave only one comment to improve this manuscript.

1. To improve readability, would it be possible to make only single energy loss profile along the layers (from -9 to 8) in Figures 3c and d?

Response: We greatly appreciate the reviewer's valuable suggestion. We do agree reviewer's comment on improving readability by making single energy loss profile along layers. We have revised Fig.3 to show the line profile of $ZO_{low\ q}$ and $ZO_{high\ q}$ between all layers, as shown in Fig. R1c. However, this lineprofile is already shown in Fig. 2d as black dots and shaded errorbars. Moreover, what we want to emphasize is the change of energy from one layer to the next layer, this information is better expressed in our last manuscript. Therefore, we think it would be better to keep the Fig. 3c-d the same as before.

Figure R1 | Vibration energy change between atomic layers. (a) HAADF image and the schematic of the structure of the $h^{-10}\text{BN}/h^{-11}\text{BN}$ interface. Each atomic layer is labeled as the number of layer away from interface (plus for $h^{-11}\text{BN}$ and minus for $h^{-10}\text{BN}$). The interface is between -1 and 1 layer. (b) EEL spectra of ZO phonons from the $h^{-10}\text{BN}$ to $h^{-11}\text{BN}$. The acquisition position of spectra are labelled in (a) by the arrows of the same color, i.e. the spectra of interface (layer No. -1 ~ 1) is shown in black, the adjacent layer

to the interface (layer No. -2 ~ -1 and No.1 ~ 2) is shown in magenta and cyan, and the bulk h-¹⁰BN (layer No. -9 ~ -5) and h-¹¹BN (layer No.5 ~ 8) is shown in red and blue. No. “n ~ n+1” label the spectra acquired between n and n+1 atomic layers. The bottom shows the fitted ZO_{high q} and ZO_{low q} modes by the multi-gaussian method. **(c)** The vibration energy profile of ZO_{low q} and ZO_{high q} along the atomic layers. The energies are extracted from the region with corresponding colors in **(a)**-**(b)**.